



**Simulating the Holocene deglaciation across a marine terminating portion of southwestern**
**Greenland in response to marine and atmospheric forcings**
Joshua K. Cuzzone[1], Nicolás E. Young[2], Mathieu Morlighem[3], Jason P. Briner[4], Nicole-Jeanne
Schlegel[5]
1University of California Los Angeles, Los Angeles, CA, USA
2Lamont-Doherty Earth Observatory, Columbia University, New York, NY, USA
3Department of Earth Sciences, Dartmouth College, Hanover, NH, USA.
4Department of Geology, University at Buffalo, Buffalo, NY, USA
5NASA Jet Propulsion Laboratory, California Institute of Technology, Pasadena, CA, USA.
*Correspondence to*: Joshua K. Cuzzone (Joshua.K.Cuzzone@jpl.nasa.gov)
**Abstract**
Numerical simulations of the Greenland Ice Sheet (GrIS) over geologic timescales can greatly
improve our knowledge of the critical factors driving GrIS demise during climatically warm
periods, which has clear relevance for better predicting GrIS behavior over the upcoming
centuries. To assess the fidelity of these modeling efforts, however, observational constraints of
past ice-sheet change are needed. Across southwestern Greenland, geologic records detail
Holocene ice retreat across both terrestrial-based and marine terminating environments, providing
an ideal opportunity to rigorously benchmark model simulations against geologic reconstructions
of ice-sheet change. Here, we present regional ice sheet modeling results using the Ice-sheet and
Sea-level System Model (ISSM) of Holocene ice sheet history across an extensive fjord region in
southwestern Greenland covering the landscape around the Kangiata Nunaata Sermia (KNS)
glacier and extending outward along the 200 km Godthåbsfjord. Our simulations, forced by novel
reconstructions of Holocene climate and recently implemented calving laws, assess the sensitivity
of ice retreat across the KNS region to atmospheric and oceanic forcing. Our simulations reveal
that the geologically reconstructed ice retreat across the bedrock landscape above sea-level in the
study area was likely driven by fluctuations in surface mass balance in response to early Holocene
warming – and likely not influenced significantly by the response of adjacent outlet glaciers to
calving and ocean-induced melting. The impact of ice calving within fjords, however, plays a
significant role by enhancing ice discharge at the terminus, leading to interior thinning up to the
ice divide that is consistent with reconstructed magnitudes of early Holocene ice thinning. Our
results, benchmarked against geologic constraints of past ice margin change, suggest that while
calving did not strongly influence Holocene ice margin migration across terrestrial portions of the
KNS forefield, it strongly impacted regional mass loss. While these results may provide an analog
to how similar fjord-dominated regions of the GrIS may respond to future warming, they also
illustrate that implementation and resolution of ice calving in paleo ice flow modeling is important
towards making more robust estimations of past ice mass change.

**1. Introduction**

Over the past few decades, the Greenland Ice Sheet (GrIS) has experienced accelerating ice mass
loss driven by increases in surface melt, runoff, and dynamic ice loss at marine terminating margins



(IMBIE, 2019). While projected mass loss from the GrIS is expected to be driven increasingly by
its surface mass balance (smb; Enderlin et al., 2014; Vizcaino et al., 2015; Goelzer et al., 2020)
and attendant meltwater runoff (Fettweis et al., 2008; Lenaerts et al., 2018), considerable
uncertainty exists regarding how oceanic forcing will influence GrIS mass loss, particularly
through ice calving processes (Goelzer et al., 2020; Choi et al., 2021). The satellite-based
observational record of GrIS change only spans a few decades making it difficult to identify and
disentangle the key drivers of GrIS mass change, and to understand over which timescales they
operate. Fortunately, geologic records detailing the retreat history of the GrIS provide an
important metric for evaluating numerical ice sheet models and help pinpoint the contributions of
various driving mechanisms to GrIS change. When combined, numerical ice sheet models and
geologic reconstructions can provide key insights into GrIS behavior in a warming climate across
centennial to millennial timescales.
The current interglaciation, the Holocene (the last 11.7 ka), is characterized by prolonged warmth
with proxy records suggesting that temperatures during the early to middle Holocene were 3±1 °C
warmer than the pre-industrial period (Briner et al., 2016; Lecavalier et al., 2017), which drove
widespread retreat of the GrIS margin at a rate of ice mass loss exceeding $20^{th}$ century values
(1900-2000 CE Young and Briner, 2015; Briner et al., 2020). Across Southwestern Greenland, a
detailed geologic record of Holocene ice-margin retreat encompassing both terrestrial and marine
terminating environments exists, providing an ideal testbed for ice sheet models to test the
sensitivity of past ice margin migration to atmospheric and marine forcings (Larsen et al., 2014;
Lesnek et al., 2020; Young et al., 2020; Young et al., 2021). Where land-based ice existed, well-
dated moraine sequences constrain ~120 km of ice retreat from the present-day coastline to just
outboard of the present-day ice margin (Lesnek et al., 2020; Young et al., 2020), and have been
shown by ice sheet models to be driven by negative smb in response to early Holocene warming
(Cuzzone et al., 2019; Downs et al., 2020; Briner et al., 2020).
Unlike the land-based portions of Southwest Greenland however, across the marine based region
covering the forefield around Kangiata Nunaata Sermia (KNS; Figure 1), it remains unknown what
drove this rapid ice margin retreat during the early Holocene (Young et al., 2021). While links
between atmospheric warming and runoff induced terminus retreat have been implicated as
reasons for the most recent historical retreat across the KNS region (Lea et al., 2014a,b), the longer
term triggers of rapid Holocene ice retreat are not constrained by the geologic data alone. Because
of the well dated chronology detailing Holocene ice retreat across this region however, ice sheet
models are well poised to address questions surrounding the scales of influence atmospheric and
oceanic forcings play on long term ice margin and mass change.
Building on recent advances in calving front dynamics in the Ice Sheet and Sea-level System
Model (ISSM; Larour et al., 2012), we use a high-resolution regional ice sheet model to investigate
the Holocene ice retreat across the KNS forefield. Our simulations build on prior ice modeling
efforts across Southwestern Greenland that were driven by novel reconstructions of past climate
(Badgeley et al., 2020; Briner et al., 2020). Where our past ice flow modeling efforts excluded ice
ocean-interactions (Briner et al., 2020), our simulations presented here take advantage of recent
implementation of physically based calving schemes in ISSM to specifically address how
Holocene ice retreat across the KNS forefield was influenced by marine and atmospheric forcing's.



Moreover, this work provides a foundation for future experiments using ISSM to simulate the
influence of ice-ocean interactions on the Holocene variability of the broader GrIS.

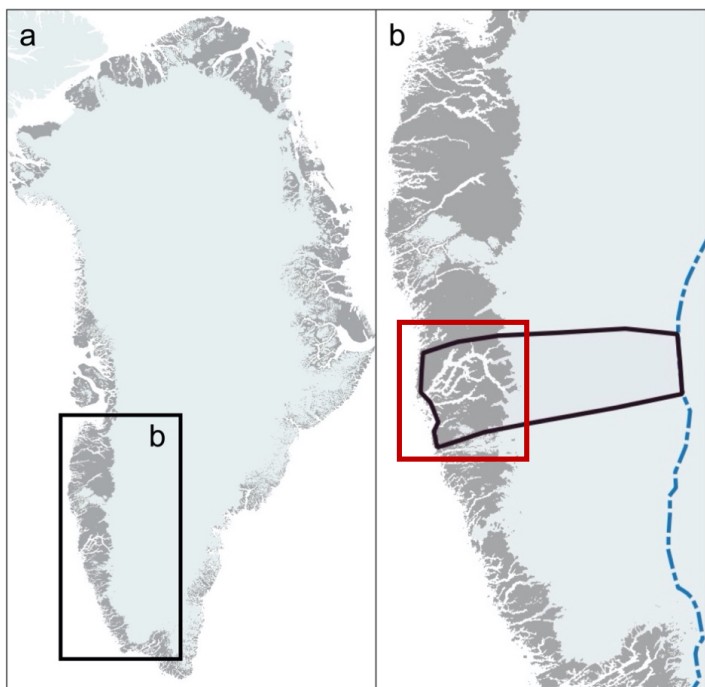

**Figure 1.** a.) The Greenland Ice Sheet. Highlighted is southwestern Greenland, where the ice model domain resides. b.) Southwestern Greenland. The ice model domain is outlined (bold black line), extending between the present-day coastline and ice divide (dashed blue line; Rignot and Mouginot, 2012). The red box corresponds to the area in figures 5, 6 and 8.

**2.    Holocene Ice Retreat across the KNS Forefield**

Decades of radiocarbon dating and, more recently, cosmogenic [10]Be dating, track the retreat of the
GrIS in the KNS region through the Holocene (Weidick et al., 2012 and references therein; Larsen
et al., 2014; Young et al., 2021). Minimum-limiting radiocarbon ages from the outer coast near
Nuuk range from ca. 11.2 to 10.6 ka BP., which is mimicked by [10]Be ages of ca. 10.7 and 10.4 ka
BP (Figure 2). Between the outer coast and the modern GrIS margin at KNS are numerous
radiocarbon and [10]Be ages that are largely indistinguishable and require rapid deglaciation of the
region spanning about a millennium (Weidick et al., 2012; Larsen et al., 2014; Young et al., 2021).
Perhaps most relevant here are [10]Be ages in the immediate KNS region from just beyond the
historical ice limit that suggest KNS had retreated within or near its current position by ca. 10.3 ka
(Young et al., 2021). Radiocarbon ages from raised marine deposits, which require ice-free
conditions, adjacent to the main KNS fjord appear slightly younger than regional [10]Be ages. These
radiocarbon ages, however, are minimum-limiting ages and an upfjord radiocarbon age of ca. 10.2
ka from a bivalve reworked by a KNS readvance requires that the main fjord deglaciated prior to
ca. 10.2 ka (Figure 2). Collectively, the radiocarbon and [10]Be ages suggest rapid and synchronous





deglaciation of both the landscape and fjord systems between the outer coast near Nuuk and the
modern margin at KNS. Lastly, [10]Be ages from slightly beyond the historical limit to the north and
south of KNS are slightly younger suggesting that these ice margins may have lagged behind ice
retreat in the immediate KNS region (Figure 2).

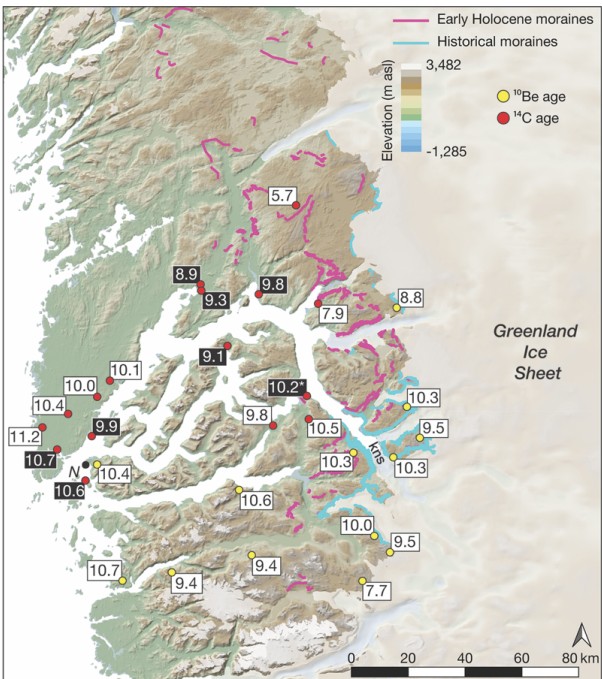

**Figure 2.** KNS region with geological constraints that track GrIS retreat in the early Holocene.
Radiocarbon ages (red circles) and [10]Be ages (yellow circles) are from Weidick et al. (2012), Larsen et al.
(2014), and Young et al. (2021). For figure clarity, we only show the mean deglaciation age at each site
(see Young et al., 2021 for full site descriptions). Radiocarbon and [10]Be across the immediate KNS region
are similar and reveal that deglaciation of the coast occurred ca. 11.2-10.7 ka and KNS had retreated near
or within its modern extent by ca. 10.3 ka. Radiocarbon ages in white text and black background are from
marine deposits and constrain the timing of retreat within the main fjord. Figure has been modified from
Young et al. (2021).


## 3.  Model description and setup



### 3.1 Ice Sheet Model



We rely on ISSM, a thermomechanical finite-element ice sheet model, to simulate the Holocene
ice history across the KNS forefield, and follow similar published model setups (Cuzzone et al,
2019; Briner et al., 2020).  The higher-order approximation of Blatter (1995) and Pattyn (2003) is
used to solve the momentum balance equations.  Our model domain centers on the KNS and
Godthåbsfjord forefield, extending from the present-day coastline, where geologic observations
show ice resided at the end of the Younger Dryas (Larsen et al., 2014; Lesnek et al., 2020) to the





present-day ice divide (Figure 1b; Rignot and Mouginot, 2012). The northern and southern
boundaries of our model domain are chosen to represent regions of minimal north-south across
boundary flow based on Holocene ice sheet simulations of southwestern Greenland (Briner et al.,
2020). Anisotropic mesh adaptation is used to create a non-uniform model mesh that varies based
upon gardients in bedrock topography from BedMachine v3 (Morlighem et al., 2017). Because
fjord width across our domain is often <5 km and high-resolution grids are necessary for capturing
grounding line dynamics (1 km; Seroussi and Morlighem, 2018), the horizontal mesh resolution
varies from 1 km in fjords and areas of high bedrock relief to 15 km where the bedrock relief is
low (Figure 3).
To capture the thermal evolution of the ice, our model uses an enthalpy formulation (Aschwanden
et al., 2012) that captures both temperate and cold ice. We impose transient air temperatures at
the surface and a constant but spatially varying geothermal heat flux at the base (Shapiro and
Ritzwoller, 2004) and our model contains only five vertical layers in order to reduce computational
load (Cuzzone et al., 2018; Cuzzone et al., 2019). In order to capture sharp thermal gradients near
the base and simulate the vertical distribution of temperature within the ice, we use quadratic finite
elements (P1xP2) along the z-axis for the vertical interpolation following Cuzzone et al. (2018).
This methodology has been successfully applied to simulate the transient behavior of the GrIS
across geologic timescales and the contemporary period (Cuzzone et al., 2019; Briner et al., 2020;
Smith-Johnson et al., 2020).
We use a linear friction law and, similar to Briner et al. (2020), we construct a spatially varying
basal friction coefficient ($k$) under areas covered by the present-day ice sheet using inverse
methods (Morlighem et al., 2010; Larour et al., 2012) that satisfies the best match between
modeled and satellite-derived surface velocities (Rignot and Mouginot, 2012):
$$\tau_b = -k^2 N v_b \tag{1}$$
where $\tau_b$ represents the basal stress, N represents the effective pressure, and $v_b$ is the magnitude
of the basal velocity. For contemporary ice-free areas, a spatially varying basal friction coefficient
is constructed to be proportional to bedrock elevation following Åkesson et al., 2018:
$$k = 100 \times \frac{\min[\max(0, z_b + 800), z_b]}{\max(z_b)} \tag{2}$$
where $z_b$ is the height of the bedrock with respect to sea level. For these parametrizations, the
friction coefficient is low within fjords and is larger over areas of high topographic relief. This
basal friction coefficient is allowed to vary through time based upon changes in the simulated basal
temperature following Cuzzone et al. (2019). As simulated basal ice temperatures decrease with
respect to present day, the friction coefficient will increase, and therefore sliding will decrease.
The opposite occurs when simulated basal temperatures are warm relative to present day. Lastly,
the ice rheology parameter $B$ is temperature-dependent, following rate factors in Cuffey and
Paterson (2010), and is initialized by solving for a present day thermal steady state and allowed to
vary during transient simulations (Cuzzone et al., 2018; 2019).


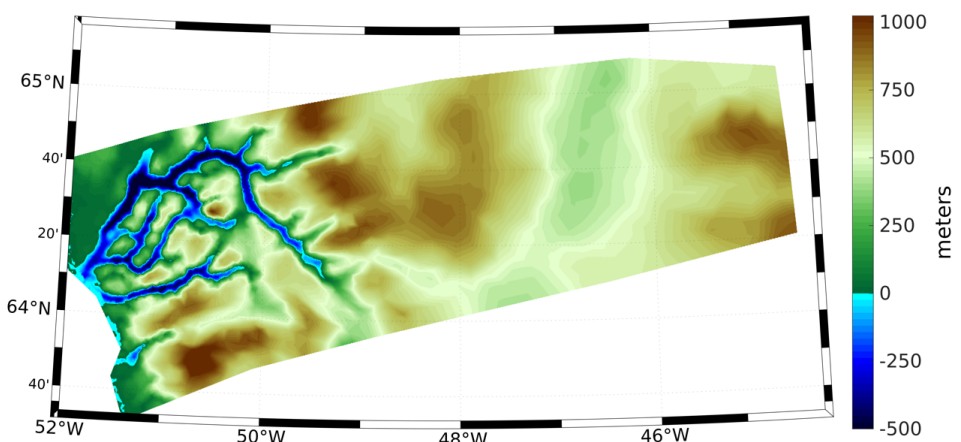

**Figure 3.** Bedrock topography for the model domain. Blue colors indicate areas that are below present-day sea level.

**3.2 Ice Front Migration and Calving**
We use the level-set method to track the motion of the ice front (Bondzio et al., 2016). The velocity
of the moving ice front is calculated as:
$$v_f = v - (c + \dot{M})\,n \tag{3}$$
where $v_f$ is the ice velocity vector, v is the ice velocity vector at the ice front, c is the calving rate,
$\dot{M}$ is the melting rate of the calving front, and $n$ is the unit normal vector pointing horizontally
outward from the calving front. For these simulations, we assume that the melting rate at the
calving front is negligible compared to the calving rate.
To simulate calving, we rely on the physically-based Von Mises stress calving (Morlighem et al.,
2016), whereby the calving rate is related to tensile stresses within the ice:
$$c = \|v\|\,\frac{\tilde{\sigma}}{\sigma_{max}} \tag{4}$$
where $\tilde{\sigma}$ is the von Mises tensile strength, $\|v\|$ is the magnitude of the horizontal ice velocity, and
$\sigma_{max}$ is the maximum stress threshold, which has separate values for grounded and floating ice.
Under this formulation, the ice front will remain stable when $\tilde{\sigma} = \sigma_{max}$, will retreat when $\tilde{\sigma} >$
$\sigma_{max}$, and will advance when $\tilde{\sigma} < \sigma_{max}$. Tensile strength measurements of ice show a range of
possible $\sigma_{max}$, ranging between 150 kPa to 3100 kPa (Petrovic 2003). For this study we choose





$\sigma_{max}$ = 600 kPa for grounded ice and 200 kPa for floating ice, which is within the ranges used by
recent studies across Greenland (Bondzio et al., 2016; Morlighem et al., 2016; Choi et al., 2020).
**3.3 Climate and Surface Mass Balance Reconstruction**
We rely on a novel gridded paleoclimate reanalysis product that reconstructs the necessary climate
variables of temperature and precipitation needed to calculate the surface mass balance history
through the Holocene (Badgeley et al., 2020). Temperature was derived from oxygen-isotope
records from eight ice cores, and five ice core accumulation records were used to reconstruct
precipitation. This reanalysis relies on a data assimilation framework that combines the
information from ice core proxies with climate-model simulations of the last deglaciation (Liu et
al., 2009; He et al., 2013) to create a spatially complete (e.g., GrIS wide) and temporally consistent
reconstruction of past temperature and precipitation. This reconstruction agrees well with
independent proxies and previously published paleoclimate reconstructions (Badgeley et al.
(2020)). For new simulations presented here, we chose two end members of reconstructed
precipitation and temperature from Badgeley et al. (2020). The high temperature reconstruction
was chosen, which has a greater magnitude of early Holocene warming, and the low temperature
scenario, which has a more muted early Holocene warming (Figure 4a). Additionally, we choose
the high and low precipitation scenarios (Figure 4b), which differ in the magnitude and timing of
peak Holocene precipitation. These reconstructions span a plausible range of temperature and
precipitation scenarios as discussed in Badgeley et al. (2020).

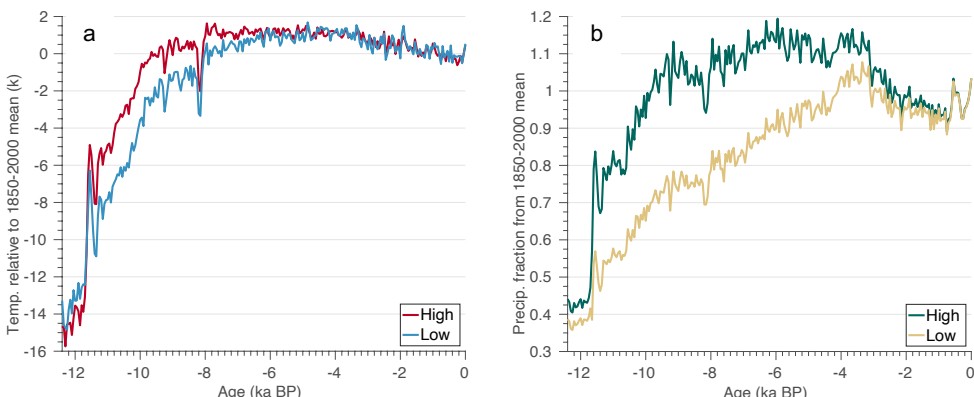

**Figure 4.** a) Area-averaged (over model domain) mean annual temperature anomaly (k) relative to the
1850-2000 mean for the High and Low temperature reconstructions from Badgeley et al. (2020). b) Area-
averaged (over model domain) mean annual precipitation as a fraction from the 1850-2000 mean for the
High and Low reconstructions from Badgeley et al. (2020).

The simulations discussed below use a combination of these forcings to address the possible role
of varying climatic conditions. Following prior work (Cuzzone et al., 2019; Briner et al., 2020),
we compute the surface mass balance over the Holocene using a positive degree day (PDD) method
(Tarasov and Peltier, 1999; Le Morzadec et al., 2015). For this scheme, snow melts first at 4.3
mm °C$^{-1}$day$^{-1}$, and the remaining positive degree days are used to melt bare ice at 8.3 °C$^{-1}$day$^{-1}$. A





lapse rate of 6 ºC/km is used to adjust the temperature of the climate forcings to ice-surface
elevation, while an allowance for the formation of superimposed ice is permitted following
Janssens and Huybrechts (2000).
**3.4 Experimental Setup**
For the reanalysis discussed in section 3.3, temperature is expressed as anomalies from the AD
1850-2000 mean, and precipitation is expressed as a fraction of the AD 1850-2000 mean (Figure
4). Following Briner et al. (2020), we apply these anomalies onto the 1850-2000 monthly mean
climatology of temperature and precipitation from Box et al. (2013) to produce the necessary
Holocene temperature and precipitation forcings:
$$T_t = \bar{T}_{(1850-2000)} + \Delta T_t \qquad (5)$$
$$P_t = \bar{P}_{(1850-2000)} \times \Delta P_t \qquad (6)$$
where $\bar{T}_{(1850-2000)}$ and $\bar{P}_{(1850-2000)}$ are the monthly mean temperature and precipitation over AD
1850-2000 from Box et al. (2013) and $\Delta T_t$ and $\Delta P_t$ are the monthly anomalies from Badgeley et al.
(2020). We perform four transient model simulations using four combinations of possible climate
scenarios shown in Table 1. For each climate scenario, we run two simulations. First, simulations
are performed where the calving parameterization is turned on (denoted as 'Calving On'). Second,
simulations are performed where the calving parameterization is turned off (denoted as 'Calving
Off'). For these simulations, we apply a temporally constant melting rate under floating ice of 40
m/yr. We also perform additional simulations discussed further in section 4.4 to assess sensitivity
to the calving maximum stress thresholds and ocean-induced melt-rates.
We initialize our regional ice-sheet model using present-day ice-surface elevation from the
Greenland Ice Mapping Project digital elevation model (Howat et al., 2014). A constant climate
from 12,400 years ago is then applied for each experiment, allowing our model to reach
equilibrium in ice volume and basal temperature, which takes 20,000 years. Since our simulations
are regional in scale, we use boundary conditions of temperature, ice velocity, and thickness from
a recent ice sheet simulation of West-Southwest Greenland (Briner et al., 2020) and impose these
as Dirichelt boundary conditions at the southern, northern, and ice-divide boundaries. These
boundary conditions are forced transiently throughout the Holocene simulations and use similar
model setups and climate forcings as discussed here. Each model is then run transiently through
time from 12,400 years ago to AD 1850 using the climatologies discussed above, and then from
1850 to 2013 we use monthly temperature and precipitation fields from Box et al. (2013). We use
an adaptive timestep, which varies between 0.02 and 0.1 years, depending on the Courant–
Friedrichs–Lewy criterion (Courant et al., 1928.



| | Temperature Scenario | Precipitation Scenario | Calving Parameterization |
|---|---|---|---|
| Experiment I | High | High | On |
| | | | Off |
| Experiment II | High | Low | On |
| | | | Off |
| Experiment III | Low | High | On |
| | | | Off |
| Experiment IV | Low | Low | On |
| | | | Off |

**Table 1.** Description of model experiments. See Figure 4 for a display of the temperature and precipitation forcings scenarios.

## 4. Results

We spin up each model as described above (section 3.4) without the ice calving parametrization turned on. Only when we begin the transient simulation through the Holocene do we turn on the ice calving parametrization for the 'Calving On' scenarios (Table 1). Our transient simulations begin 12,400 years ago with the ice margin residing along the present-day coastline for all experiments, which is approximately consistent with where geologic constraints place the ice margin at that time (Young et al, 2021 and references therein).

### 4.1 Simulated Deglaciation

First, we assess how our simulated deglaciation compares with geologic reconstructions of ice sheet change in the KNS region. Geological constraints outlined above reveal that ice retreated across the KNS forefield rapidly in the early Holocene. While relatively little direct information exists detailing ice retreat within the fjords, the terrestrial portion of our domain (i.e., the inter-fjord bedrock landscape) became ice-free between ~11.2 ka and 9.5 ka as ice retreated from the modern coastline towards, and eventually surpassing, what is now the modern ice margin.

To compare against the geologic constraints, we determine when in time portions of our model domain become ice free (Figure 5). Since ice can readvance over areas that had been deglaciated during our simulations, we take the youngest age from which locations in our simulations became ice free. Our simulations illustrate clear differences in the timing of deglaciation across terrestrial surfaces above sea-level and within the fjords. For the high and low temperature scenarios, terrestrial surfaces deglaciate up to a few millennia earlier than the adjacent fjords. This difference in timing between the fjords and terrestrial surfaces is perhaps unsurprising given how fjord systems act as conduits draining the ice interior. This persistence of ice extent within the fjords despite elevated warming experienced during early to middle Holocene illustrates the role of ice dynamics, which is explored further in section 4.3.

For the high and low temperature scenarios, there is little difference between the age of deglaciation on terrestrial surfaces for simulations that allow (Figures 5a and 5b; Figures 6a and 6b) and do not allow calving (Figures 5d and 5e; Figures 6d and 6e). In contrast, deglaciation of terrestrial surfaces occurs later in Holocene for the simulations using the high precipitation





scenario than for those simulations using the low precipitation scenario. For simulations using the
high temperature scenario, these differences are up to 500 years (Figure 5). For the low temperature
scenarios, terrestrial surfaces deglaciate up to 1000 years later for simulations using the high
precipitation forcing (Figure 6).

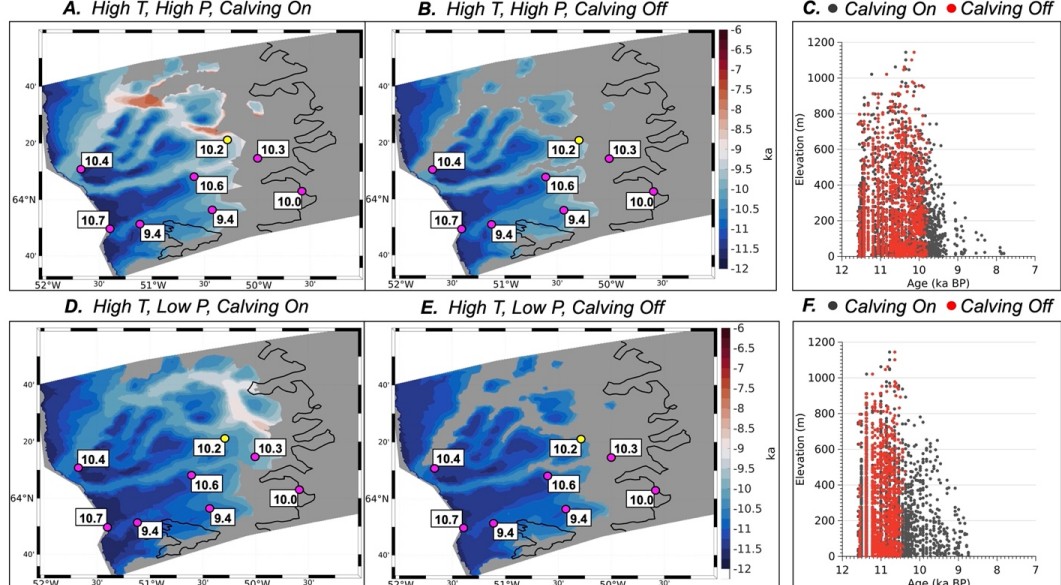

**Figure 5**. Map of simulated deglaciation ages for High temperature scenarios with A.) High precipitation, Calving On. B.) High precipitation, Calving Off. D.) Low precipitation, Calving On, and E.) Low precipitation – Calving off. Gray mask is the simulated ice extent at present day and the black line denotes the actual present day ice extent (Rignot and Mouginot, 2012). Magenta circles are the best estimate of the timing of deglaciation at that point based on [10]Be surface exposure ages in thousands of years ago and the yellow dot shows minimum limiting radiocarbon age (Young et al., 2021). Scatter plot of simulated deglaciation age (above sea level) versus bedrock elevation for C) High temperature, high precipitation, and F) High temperature, low precipitation. Red dots are from simulations without calving and black dots are for simulations with calving.

The larger sensitivity to the precipitation reconstruction on the timing of deglaciation for the lower
temperature scenario versus the high temperature scenario is similar to the results of Briner et al.
(2020). Across deglaciated regions where the surface mass balance dictates ice margin migration,
increased precipitation modulates the temperature driven retreat in the early Holocene, particularly
for simulations with colder climates (Briner et al., 2020; Downs et al., 2020). As discussed above,
since calving does not appear to significantly influence terrestrial ice retreat across this region,
SMB may be the primary driver of ice retreat across terrestrial surfaces within our model domain
from which the majority of geologic constraints on past ice retreat are present. It is important to
note that, while calving does not seem to play a significant role in the retreat of ice across terrestrial





surfaces, simulations that allow calving have a more reduced ice extent (gray mask) at the end of
each simulation, which may indicate that calving limits ice front readvance within the fjord.

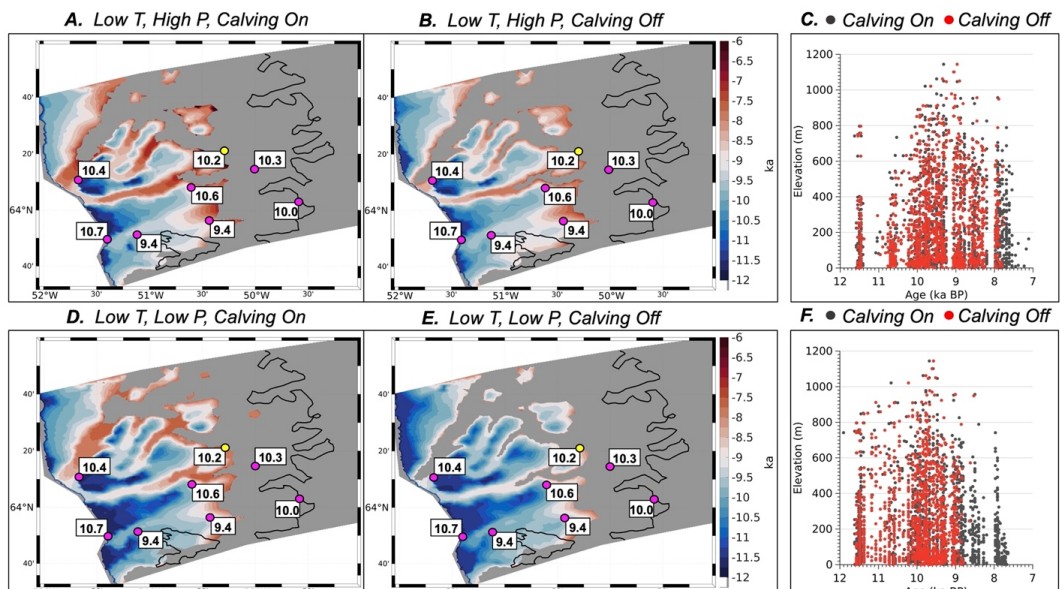

**Figure 6**. Map of simulated deglaciation ages for Low temperature scenarios with A.) High precipitation,
Calving On. B.) High precipitation, Calving Off. D.) Low precipitation, Calving On, and E.) Low
precipitation – Calving off. Gray mask is the simulated ice extent at present day and the black line denotes
the actual present day ice extent (Rignot and Mouginot, 2012). Magenta circles are the best estimate of
the timing of deglaciation at that point based on [10]Be surface exposure ages in thousands of years ago and
the yellow dot shows a minimum limiting radiocarbon age that requires ice free conditions in the fjord at
that time (Weidick et al., 2012; Young et al., 2021). Scatter plot of simulated deglaciation age (above sea
level) versus bedrock elevation for C) Low temperature, high precipitation, and F) Low temperature, low
precipitation. Red dots are from simulations without calving and black dots are for simulations with
calving.

The manner in which deglaciation occurs on terrestrial surfaces can be an important factor in
determining the pace and magnitude of the ice margin response to warming. Geologic archives
constraining ice retreat across the KNS forefield span an elevational range of 1300 m, yet, no
elevational dependence on the age of deglaciation is evident (Larsen et al., 2014; Young et al.,
2021). This could indicate large scale ice margin retreat in response to rapid ice surface lowering,
but certainly precludes scenarios where ice surface lowering occurred slowly exposing high
elevation sites well before low elevation sites. To compare our simulated deglaciation history as
a function of elevation against the geologic data, we plot the simulated age of deglaciation against
elevation, and restrict our datapoints to terrestrial surfaces above sea level (Figure 5c and 5f; Figure
6c and 6f). In general, our simulations agree with the geologic data indicating that there was no
elevational dependence on the age of deglaciation; if there were any indication of an elevation
dependence on the age of deglaciation, we would observe that high elevation sites would become
ice free first, followed by low elevation sites. Instead, all of the plots show that deglaciation
happens simultaneously at discrete time intervals across all elevation bands, indicating that ice



surface lowering was rapid and coincident with ice margin pullback.  These elevation-time
diagrams also highlight how the higher precipitation scenarios have later mean deglaciation ages
across terrestrial surfaces (Figure 5c and 6c) than corresponding simulations using the low
precipitation scenario (Figure 5f and 6f). We also note that for simulations where calving is turned
off (red dots), ice retreat appears to stop earlier than for those simulations with calving turned on
(black dots).  This occurs because the simulations without calving experience a larger late
Holocene ice readvance than those simulations where calving is turned on (black dots).  As a
consequence of this, model grid points that would have otherwise deglaciated prior to the
readvance are overrun with ice and therefore are not marked as deglaciated in the simulation.
Lastly, each of our experiments end with a simulated present-day ice extent that is beyond
(westward of) the actual present-day ice extent (Figure 5 and 6).  Yet, the simulated ice-margin
position in the fjords is less extensive for all experiments where calving is permitted.  Those
experiments that allow calving and used the high temperature scenario (Figure 5a and 5d) simulate
a present-day ice extent that is closer to the observed present-day margin when compared to
simulations using the low temperature forcing (Figure 6a and 6d).

**4.2 Ice mass evolution and minimum ice extent**
Broadly, scenarios that allow calving undergo greater ice mass loss than those simulations where
calving is not allowed (Figure 7; black lines). The differences in simulated ice mass also vary
depending on the climate scenarios used.  For example, during early Holocene warming (12 ka -
8 ka), simulations that allow calving and use the high temperature scenarios (Figure 7a, b)
experience ice mass loss, while simulations that do not allow calving experience a period of ice
mass stability (Figure 7a, b; dashed red line), which is more prolonged in the simulation using the
high precipitation scenario (Figure 7a).
For the simulations using the low temperature scenario (Figure 7c, d), initial ice mass loss is
interrupted by brief increases in ice mass during the early Holocene (between 11 ka-10 ka).  This
increase in ice mass occurs for both scenarios with and without calving (Figure 7c, d; black and
dashed red line), although the simulations without calving experience larger increases in ice mass
during this period.  Accordingly, the low temperature simulation with higher precipitation (Figure
7c) experiences larger ice mass gain than the simulation using the low precipitation scenario
(Figure 7d).  During this interval, precipitation is approximately 20-30% more for the high
precipitation scenario during the early Holocene than the low precipitation scenario.  Much of this
mass gain is due to ice thickening over the interior of the model domain, where despite early
Holocene warming, colder temperatures (at higher elevations on the ice sheet) support snowfall
(see section 4.3).
Throughout the remainder of the Holocene, the evolution of ice mass for experiments using the
high temperature scenario (Figure 7a, b) differ from those simulations using the low temperature
scenario (Figure 7c, d).  Simulations using the high temperature scenario (Figure 7a, b) reach a
minimum ice volume between 7.6-7.2 ka.  For the simulation using the high precipitation scenario,
ice mass increases slightly following this minimum, and remains generally stable throughout the
remainder of the Holocene (Figure 7a), whereas the simulation using the low precipitation scenario
experiences large ice mass gain following this minimum, with steady growth occurring throughout

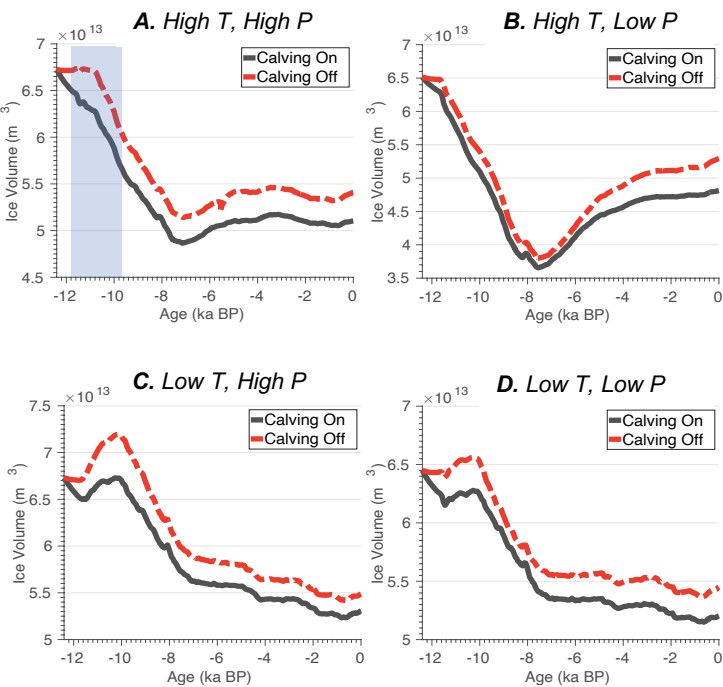

**Figure 7.** Holocene ice volume (x10$^{13}$ m$^3$) evolution for each model experiment. Refer to Table 1 for a summary of the climate forcings used in each experiment. Black lines denote those simulations with the calving parametrization turned on. Dashed red lines denote those simulations with the calving parametrization turned off. The vertical blue bar above marks a time period (12 ka – 10 ka) used for analysis presented in Figure 8 and 9.

the remainder of the simulation (Figure 7b). It is important to note, however, that for the high
temperature scenarios, this ice mass gain is more muted for simulations that allow calving. In
contrast, the simulations using the low temperature scenario (Figure 7c, d) lose the majority of ice
mass by 8-7 ka, with ice mass loss either continuing through the Holocene (Figure 7c) or remaining
relatively stable before reaching a minimum at 0.6-0.4 ka (Figure 7d).
Regional relative sea-level records reveal that sea level fell below modern between 4-3 ka, before
rising towards modern values (Long et al., 2011), interpreted to represent the re-loading of the
Earth's crust as the GrIS readvanced during the late Holocene following a mid-Holocene
minimum. In addition, radiocarbon-dated lake sediments from southwestern Greenland suggest
that this sector of the GrIS likely achieved its minimum extent after ca. 5 ka, and that eastwards
retreat of the ice margin was likely minimal (Larsen et al., 2015; Young and Briner, 2015; Lesnek
et al., 2020; Young et al., 2021). Although no direct geological constraints on the minimum GrIS
ice extent during the Holocene exist, available constraints suggest that the large-scale ice margin
retreat inboard of the present-day extent as simulated by some ice sheet models in this sector (20-
40 km; Tarasov and Peltier, 2002; Lecavalier et al., 2014) is likely too extreme. Relying on these



geologic constraints, we can crudely assess the temporal and spatial patterns of the simulated ice
mass and minimum extent.
None of our simulations accurately capture the exact timing of the GrIS minimum in the KNS
region, but some simulations are likely better representations than others. Simulations using the
high temperature scenario (Figure 7a, b) achieve an ice mass minimum prior to 5 ka followed by
ice regrowth. The high temperature-low precipitation scenario depicts an extreme GrIS minimum
followed by significant regrowth. While of the overall pattern of a GrIS minimum followed be

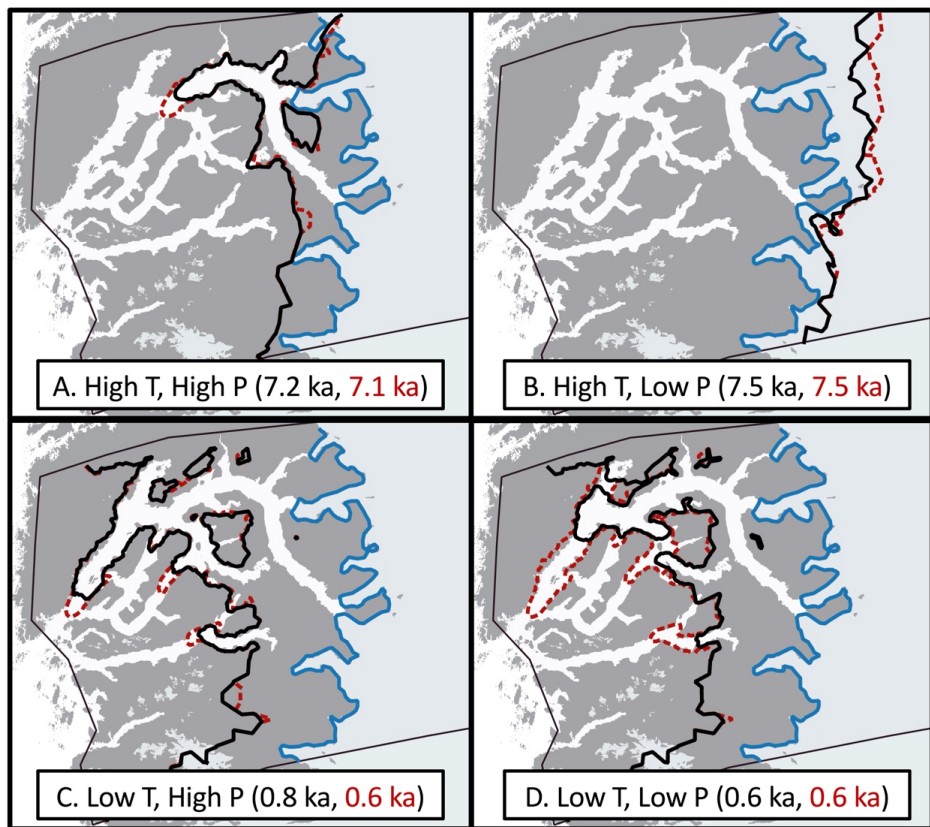

Figure 8. Age of minimum ice extent for each simulation (black text: simulations with calving, red text: simulations without calving). The black line denotes the minimum ice extent for simulations with calving. The dashed red line denotes the minimum ice extent for simulations without calving. The present-day ice extent is shown as the blue line.

regrowth is consistent with the geologic record, the magnitude of simulated change is likely
inconsistent with geological records, pointing to a rather modest GrIS minimum; although we do
acknowledge that minimal ice retreat as constrained by the geologic record does not necessarily
equate to muted mass loss. In contrast, the high temperature-high precipitation experiment depicts
an ice-mass minimum that is likely too early, but the magnitude of this minimum is less (Figure
7A). Moreover, ice regrowth following this minimum is restricted with only modest change



occurring over the last 6 kyr (Figure 7A). Although this simulated minimum is likely too early, a simulated ice mass that undergoes minimal change over the last ~6 kyr is broadly consistent with the geological record that depicts a minimum closer to ca. 4-3 ka, but where the GrIS margin likely did not undergo significant change between ca. 7-3 ka (Young et al., 2021). Both low temperature scenarios are inconsistent with the geological record as both show continued ice mass loss through the Holocene. Although it is possible, but unlikely, that continued ice loss through the Holocene could still be achieved if the ice margin retreated inland followed by a readvance toward its present position, mass loss through the Holocene is inconsistent with relative sea-level records.

The minimum ice margin extent achieved in our simulations is shown in Figure 8. For the high temperature scenarios (Figure 8a, b), the simulated minimum ice extent is either just outboard of the present-day ice margin (Figure 8a; high precipitation) or inboard of the present-day ice margin (Figure 8b; low precipitation). Because the geologic evidence supports that the Holocene ice extent minimum was close to and perhaps slightly inboard of the present-day ice margin (Young et al., 2021), both simulations are broadly consistent with the geological record. But, again, the high temperature – high precipitation scenario depicts significant ice regrowth resulting in a present-day ice margin significantly more extended than modern (Figure 5).

**4.3 Early Holocene Thinning**

Figures 9 and 10 show the simulated ice elevation changes for the time period between 12 ka to 10 ka for each experiment (highlighted in Figure 7a as the light blue vertical bar). During this time period, widespread early Holocene warming drove increased ice melt along the margin of the model domain. This pervasive thinning along the margin is captured in all model experiments (Figure 9 and 10), although the amplitude of ice thinning is greatest for the experiments using the high temperature scenario (Figure 9). Across all experiments, inland thickening occurs, however, the magnitude of interior thickening is not solely influenced by the SMB, but is also influenced by calving. For our experiments that allow calving, interior thickening is reduced and ultimately influences the trend and magnitude of changes in simulated ice volume; simulations that allow calving either experience increased ice mass loss (Figure 7a, b) or more muted ice mass gain between 12 ka and 10 ka (Figure 7c, d). Additionally, the spatial pattern of elevation changes shows that marginal thinning propagates farther upstream and into the ice sheet interior for simulations that allow ice calving. This relationship continues throughout the remainder of the Holocene, as experiments with calving either result in more mass loss than simulations without calving, or more muted ice mass gain (see Figure 7). These variations in simulated Holocene ice mass and ice surface elevation change can be linked to the influence ice calving has on ice front position and stability, and ultimately the rate at which ice can flux through the fjord system. During the time period of 12 ka to 10 ka, ice velocity differences for simulations with and without calving are in excess of 200 m/yr along many fjords within the KNS region (Figure 11). Calving at the ice front leads to increases in ice velocity within outlets across the model domain, thereby promoting increased mass flux and transport from the ice interior to the margin. Thus, even though the large-scale ice margin migration across our model domain is relatively insensitive to calving, the overall mass budget and surface profile of the ice is strongly influenced by calving.



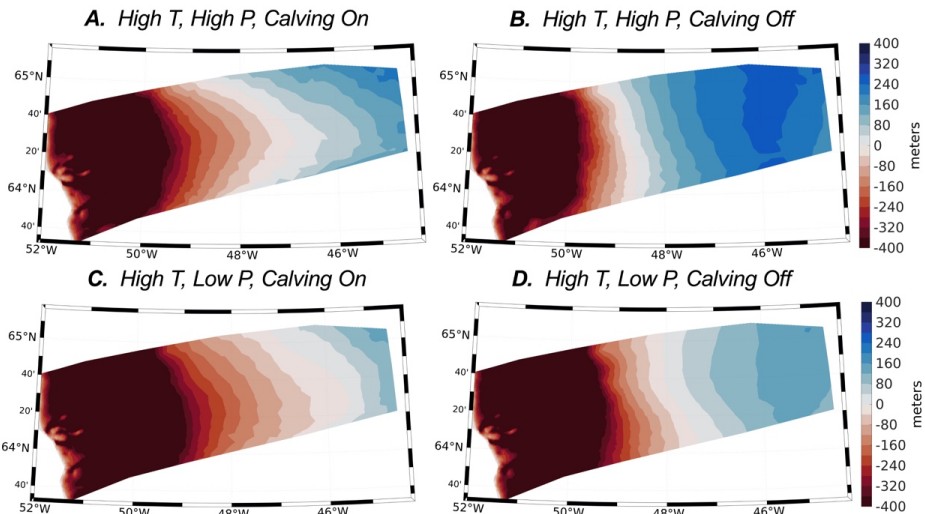

**Figure 9.** Simulated elevation changes (in meters) during period 12ka – 10ka shown for experiments using the high temperature forcing.

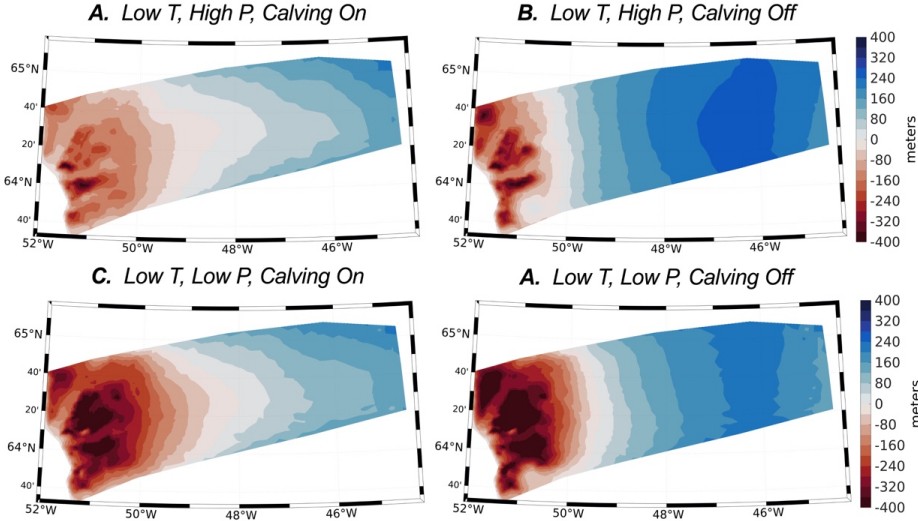

**Figure 10.** Simulated elevation changes during period 12ka-10ka shown for experiments using the low temperature forcing.


Reconstructions of Holocene ice thickness across the GrIS are limited, but ice-core records provide
a long-term perspective of dynamic changes in GrIS elevation at locations at or near the ice divide
(Vinther et al., 2009; Lecavalier et al., 2017). For example, some locations experienced more rapid
thinning in response to Holocene warming (i.e. Camp Century, Dye 3) while other locations


experienced more muted ice elevation changes (i.e. GRIP, NGRIP). A feature of many of these
records, however, is the presence of early Holocene thickening, potentially triggered by increased
snowfall at higher elevation sites as the climate warmed or by elevation-mass balance feedbacks
driven by isostatic uplift (Vinther et al., 2009). Across all model experiments, our simulated timing
of inland thickening coincides with thickening experienced at high elevation ice core locations
(Vinther et al., 2009). The magnitude of early Holocene thickening from ice core records (Vinther
et al., 2009; 11.7 ka-10 ka) is on the order of 30 – 70 meters. Therefore, our simulations that allow
calving display inland thickening (<120 m) over the time interval 12 ka – 10 ka that is more
consistent with thickening estimated from ice cores than simulations with no calving (>200m).

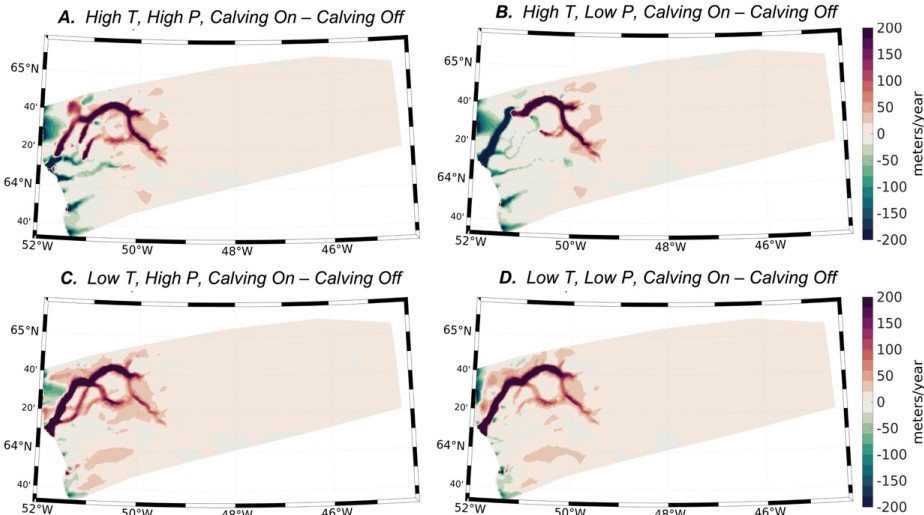

**Figure 11.** Simulated ice velocity differences between simulations with and without calving for each experiment over the time period 12 ka to 10 ka. Red colors denote an increase in ice velocity for simulations with calving relative to simulations without calving. Green colors denote a decrease in ice velocity for simulations with calving relative to simulations without calving.

**4.4 Sensitivity to marine forcing**
Experiments on the tensile strength of ice show that stress thresholds can vary between 150 kPa
and 3100 kPa (Petrovic, J., 2003), with modeling experiments on Jakobshavn glacier suggesting
that the stress threshold for grounded ice can vary between 100 kPa to 4 MPa seasonally (Bondzio
et al., 2017). Here, our grounded ice stress threshold is set to 600 kPa. Because our model setup
incurs large computational expense, we did not perform a full uncertainty analysis on these
parameterizations. Due to the nature of modeled variation in calibrated stress thresholds across
Greenland (Choi et al., 2021), however, we ran a small set of experiments where we set the calving
stress threshold on grounded ice to 1 MPa. We performed the transient simulations on the high
and low temperature scenario cases using the high precipitation forcing (see Table 1).
Additionally, we ran a set of experiments where the basal melt rate on floating ice was set to 120
m/yr. Figure 12 shows the simulated ice volumes for these experiments where the calving stress
threshold of grounded ice and basal melt rate on floating ice were changed. These experiments



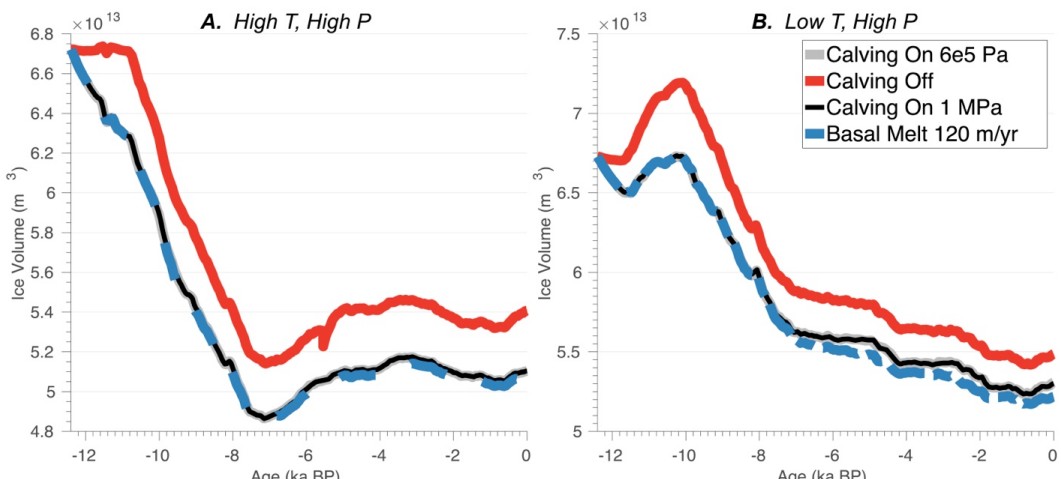

**Figure 12.** Sensitivity to the calving stress threshold for grounded ice and basal melt rates on floating ice. Red line: ice volume evolution for the simulations where the calving parameterization was turned off. Black line: ice volume evolution for the simulations where the calving stress threshold for grounded ice is 1 MPa. Gray line: ice volume evolution for the simulations where the calving stress threshold for grounded ice is 6 kPa. Dashed blue line: ice volume evolution for the simulations where the basal melt rate on floating ice was set to 120 m/yr (with calving stress threshold for grounded ice = 600 kPa).

reveal that adjusting the stress threshold from 600 kPa to 1 MPa has no effect on the evolution of
the simulated ice volume. Accordingly, increasing the basal melt rate on floating ice has minimal
effect on the simulated ice volume (Figure 12). Ice only begins to float in our experiments when
the ice front retreats into the deeper fjord bathymetry within the KNS forefield (see Figure 3), and
therefore submarine melting of floating ice seems to have limited influence on simulated ice mass
changes.
**5. Discussion**
**5.1 Terrestrial vs. Marine ice retreat**
Southwestern Greenland hosts a rich record of geologic constraints on past ice-sheet change
(Lesnek et al., 2020). Whereas a series of well-defined moraines constrain early Holocene ice
retreat across portions of southwestern Greenland dominated by terrestrial ice-margin settings
(Larsen et al., 2014; Lesnek et al., 2020; Young et al., 2020; Young et al., 2021), the Kapisigdlit
moraine system (Figure 2: early Holocene moraines) near the present-day ice margin is the only
regionally traceable moraine within the marine-dominated KNS forefield. Instead, ice-margin
retreat across the KNS forefield is constrained primarily by minimum limiting radiocarbon ages
and [10]Be surface exposure ages on deglaciated bedrock surfaces and glacial erratics (Larsen et al.,
2014; Young et al., 2021). The lack of moraine systems between coast and ice is consistent with
the relatively high rate of deglaciation estimated from the existing chronology. These
chronological constraints detail widespread and rapid retreat of the ice margin across this domain
in the early Holocene, with the ice margin retreating from the coastline around 12 ka to near the
present-day ice margin between 10-9.5 ka (Young et al., 2021). This relatively rapid retreat based



on geological observation is consistent with the lack of elevation-age relationship in our
simulations of ice margin change.
While the rapid retreat of the terrestrial ice margin is well constrained, how ice retreated up the
fjords is less certain. Our simulations depict a pattern of ice retreat across the landscape that was
largely independent of ice retreat within fjords, which lagged by 0.5 – 2 ka. For our simulations,
scenarios using the same climate forcing show little difference (<1 ka) in the simulated age of ice
retreat on terrestrial ice margins regardless of whether calving is allowed (Figures 5 and 6). The
timing and rate of Holocene ice retreat across terrestrial portions of the KNS forefield, however,
is strongly dependent on the climate forcing used, and ultimately the SMB. The earliest ice retreat
occurs in simulations that use the high temperature scenario. Ice retreat occurs later in simulations
that use the low temperature scenario, which has a delay in the timing and magnitude of Holocene
warming (Figure 4). The pace and magnitude of ice retreat is shown to be modulated depending
on precipitation similar to the findings of Briner et al. (2020) and Downs et al. (2020), with delayed
and less rapid ice retreat in scenarios with higher precipitation (Figures 5a and 6a). These results
point to the strong influence that climate and, in particular, precipitation can have on modulating
the temperature driven response of Holocene deglaciation. Indeed, select proxy records suggest
that southwestern Greenland may have experienced a prolonged period of anomalously high
snowfall in the early Holocene, perhaps driven by increased moisture flux from Baffin Bay and
the Labrador Sea as sea-ice extent declined (Thomas et al., 2016). Ice flow modeling across
southwestern Greenland has also revealed that elevated precipitation may have accompanied early
Holocene warming (Downs et al., 2020). And recent evidence from a shallow ice core in western
Greenland reveal that significant variations in precipitation occurred in the last two thousand years
across the margins of the GrIS, whereas this variability is not present in ice core data at the interior
of the GrIS (Osman et al., 2021). Because current climate reconstructions employed in
paleoclimate ice flow modeling use either simple scaling approaches to reconstruct past climate or
rely on information from interior ice cores, large hydroclimate shifts that occur at the ice sheet
margin may not be captured (Badgeley et al., 2020). Continued progress in reconstructing past
climate will certainly improve our understanding of climatic controls on the long-term response of
the GrIS.
In general, simulations using the high temperature scenario experience terrestrial ice retreat that
occurs during 11.5 ka to 9 ka, a time window consistent with the geological record of ice-margin
change in our domain (Larsen et al., 2014; Young et al., 2021). Simulations using the low
temperature scenario reveal terrestrial ice retreat also beginning ca. 11.5 ka, but deglaciation of
our model domain continues until ~7.5 ka. In comparison, geological constraints suggest that by
~10.3-9 ka BP the ice margin in the immediate KNS region had already retreated back to, and
likely behind, what is the present-day ice margin (Young et al., 2021). Ice surface lowering is
captured in all of our simulations, which indicate that on terrestrial surfaces ice retreat was
synchronous across low and high elevations. While ice calving does not seem to significantly
influence the rate and timing of ice retreat across terrestrial portions of our domain, late Holocene
ice readvance within fjords is more restricted in those simulations that use the calving
parametrization. Accordingly, flowband modeling of KNS over the period historical period of
1761 to 2012 suggests that marine ice-front retreat was primarily influenced by atmospheric
warming and runoff, which helped to trigger ice front retreat via a crevasse-depth calving criterion,
with submarine melting only playing a minor role on historical retreat (Lea et al., 2014; Lea et al.,



2014). These results do suggest though that climate anomalies were the main driver of historical
ice terminus advance and retreat across KNS (Lea et al, 2014), with our results suggesting that the
longer-term Holocene ice terminus position was also primarily driven by atmospheric warming
and not through oceanic melting.
**5.2 Role of ice calving on mass transport**
Mass transport from the ice sheet interior to the margin plays an important role in ice sheet mass
change and ultimately its contribution to sea-level rise. Contemporary satellite-derived
measurements show inland thickening at high elevations across portions of the GrIS in response
to increased snowfall despite pervasive thinning at lower elevations (Smith et al., 2020). Although
the response of marine terminating portions of the GrIS and how it translates to interior ice mass
loss can be spatially varying (Williams et al., 2021), thinning at the ice margin due to dynamic- or
SMB-driven ice loss can elicit changes in driving stresses, which can propagate up glacier and into
the interior of the ice sheet (Price et al., 2008; Schlegel et al., 2013; Csatho et al., 2014; Felikson
et al., 2020; Williams et al., 2021).
While there is no apparent influence of ice calving on the Holocene ice retreat across the KNS
forefield over terrestrial surfaces, our simulations show that ice calving has a significant influence
on the evolution of the total ice volume. Ultimately, ice calving leads to an acceleration of ice
flow within outlet glaciers that promotes local ice thinning first, followed by propagation of this
thinning into the interior of the ice sheet, consistent with contemporary observations (Csatho et
al., 2014; Williams et al., 2021). Initially, interior ice surface elevation increases in our
simulations, with simulations that allow calving being more consistent with ice-core derived
surface height records (Vinther et al., 2009). Surface lowering near the ice margin driven by a
more negative SMB in response to early Holocene warming causes the ice surface slopes to steepen
in our domain, increasing driving stresses and mass transport. This helps drive interior ice
thinning, as shown by elevation changes in simulations that allow ice calving (Figures 9 and 10),
leading to increased ice flux at the margin through the ice streams (Figure 11). This increased
mass transport helps limit thinning within outlet glaciers, and where terrestrial locations of our
domain become ice free early in the Holocene, ice front retreat within the fjords lag (Figure 5 and
593    6).
Our results suggest that, while calving did not play a significant role in the observed Holocene ice
retreat across the KNS forefield, it played an important role on the overall ice mass change across
our model domain. These results highlight that the inclusion of physically based ice calving
parameterizations is an important step towards modeling the fidelity of simulated ice mass change
across paleoclimate timescales. However, the choice of which ice calving parameterization is best
suited to Greenland over such timescales is still not well constrained (Goelzer et al., 2017). It
remains important though, that models maintain high enough spatial resolution in order to capture
fjord environments, associated bathymetry, and ultimately ice calving and grounding line
migrations over paleoclimate timescales (Cuzzone et al., 2019) as the model resolution can impact
simulated ice discharge significantly (Rückamp et al., 2020; Ashwanden et al., 2019).
**5.3 Model limitations**





Fjord systems in Greenland are typically <5 km in width, making it necessary to implement high-
resolution meshes to resolve these features. Our model setup relies on a high-resolution mesh that
is able to capture the fjord geometry within the KNS forefield, making it possible to simulate
grounding line migration and calving. The calving parameterization used does ignore frontal
melting at the grounded ice front. Frontal melt at the base of a calving face has been shown to
induce undercutting of the ice front, and greatly increases calving rates (O'leary and
Christofferson, 2013). For the present day, many of southwestern Greenland's marine terminating
glaciers are not strongly influenced by undercutting (Wood et al., 2021), but this may have been
different as ice retreated up fjord to its present-day location through the Holocene. Due to a lack
of constraints on the long term subsurface ocean thermal forcing needed to implement undercutting
in our simulations, we opted to disregard this. To circumvent this shortcoming, we set our calving
stress threshold on grounded ice to a number (600 kPa) that is on the lower end of measured tensile
stresses of ice (Petrovic, 2003). Since there was no discernable difference in our simulated ice
mass change when a higher calving stress threshold of grounded ice was uses (1 MPa), we
cautiously assume that implementation of undercutting would have a negligible effect on the
calving rates and overall Holocene mass change and ice retreat across our domain.
At the time of this work, ISSM is undergoing improvements and new implementation of solid earth
and sea-level feedbacks. While we did not include time dependent forcings (e.g. Caron et al.,
2018) that account for relative sea-level change as we have in prior research (Cuzzone et al., 2019;
Briner et al., 2020), future simulations using ISSM will explore the influence of coupled solid
Earth-ice feedbacks on ice retreat. Recent ice sheet modeling (Kajanto et al., 2020) showed that
the Holocene retreat of Jakobshavn Isbræ was insensitive to relative sea-level (RSL) variations, as
RSL changes were small in comparison to fjord depth. RSL changes during the Holocene across
this domain were relatively small (~60-100 meters at 12.4 ka and decreasing through the Holocene;
Caron et al., 2018) compared to ford depths. Given that ice calving did not seem to largely
influence terrestrial ice retreat, we only expect that inclusion of Holocene RSL changes may have
influenced ice front retreat that migrated into deeper waters where floating extensions of the ice
front could occur. However, in our sensitivity tests, basal melting on floating ice plays a trivial
role in total ice volume changes (Figure 11) as most of the ice within fjords is grounded during the
Holocene retreat.
**6. Conclusions**
Understanding how climate, calving, and marine processes contribute to ice sheet change across
paleoclimate timescales is challenging. Models with lower resolution meshes are typically favored
to ensure computational needs are satisfied. This ultimately leads to poor representation of
bedrock topography (Cuzzone et al., 2019; Jones et al., 2021) and grounding line migration
(Seroussi et al., 2018) that control ice flow (i.e., fjords), making the assessment of how ice calving
influences large scale ice margin change difficult. Moreover, while ice core records provide
snapshots of a changing climate at the ice-sheet interior, there remain a relative lack of
paleoclimate records from the ice sheet margin of sufficient resolution that can be easily
incorporated into an ice sheet model's climate forcing.
Here, we presented results from a high-resolution 3D thermomechanical regional ice sheet model
that evaluated controls on the behavior of the southwestern GrIS during the Holocene in the

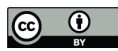



vicinity of the KNS forefield, an area with extensive geologic constraints on past ice margin
change. Experiments were driven by novel reconstructions of Holocene climate (Badgeley et al.,
2020) and included a physically based ice calving parametrization (Morlighem et al., 2016).
Our modeling results shed light on the well constrained observations of Holocene ice retreat across
the KNS forefield. Our simulations agree well with observations that ice retreat on terrestrial
bedrock surfaces occurred rapidly between 11.5 ka to 9.5 ka in response to early Holocene
warming. Variations in the timing and magnitude of ice retreat on terrestrial bedrock surfaces
across this region are insensitive to calving within the fjords that intersect this landscape, and is
instead more sensitive to the SMB with warmer climate reconstructions providing the best fit
between the modeled and observed ice retreat. Calving across this domain plays a significant role
in the simulated Holocene ice volume change. Acting as conduits for mass transport and ice flux,
ice velocity within the fjords in the KNS forefield increases when the ice front is allowed to calve.
Calving helps promote further ice mass transport from the interior of the domain to the ice front.
This helps to thicken ice within the fjords, allowing the ice front to persist longer than adjacent
terrestrial margins similar to the ice response simulated for the Holocene retreat of Jakobshavn
Isbræ (Kajanto et al., 2020). However, as all simulations depict contemporary ice extent that is
too extensive, uncertainties in the reconstruction of past climate and model parametric
uncertainties ultimately contribute to misfits that are difficult to quantify given our
computationally expensive model setup. Future paleoclimate ice flow modelling with ISSM will
aim to take advantage of recent advances in statistical emulation (e.g., Edwards et al., 2021) to
better quantify the influence of model parametric uncertainty on simulated Holocene ice retreat.
Geologic archives serve an important role in our understanding of glacier and ice sheet response
to climate change. In turn, ice sheet modeling can help improve our understanding of the climatic
and ice dynamical factors that led to ice sheet changes preserved by the geologic record. Our
modeling results present an exploration of the factors that may have contributed to the observed
pattern of Holocene ice retreat across the KNS forefield, echoing that model–data comparisons
between ice sheet models and geologic reconstructions can help improve our understanding of
long-term ice sheet sensitivity to climatic and dynamic forcing mechanisms.
**Data Availability**
The simulations performed for this paper made use of the open-source Ice Sheet System Model
(ISSM) version 4.19 and are publicly available at https://issm.jpl.nasa.gov/ (Larour et al., 2012).

**Acknowledgements**
Funding for this study was provided by the National Science Foundation Grant ARC
no. 2105960 to JC and no. 1503959 to NEY.

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
