# Peer review of "Simulating the Holocene deglaciation across a marine terminating portion of southwestern Greenland in response to marine and atmospheric forcings"

_The Cryosphere, 2022_

## Referee Comment (RC1)

**Review of Cuzzone et al., TCD**

In this paper Cuzzone et al. present a series of simulations of the Nuup Kangerlua (Godthåbsfjord) region, driven primarily by novel palaeoclimate simulations of temperature and precipitation. Within these, they test high/low temperature/precipitation scenarios and scenarios for each where a level-set calving criterion is turned on/off. Their results show surface mass balance to exert a strong control on rapid retreat (consistent with geological evidence), while calving primarily exerts a strong control on ice dynamics (and therefore the evolution of total domain volume that is arguably more important). While their simulations are not able to recreate the contemporary ice margin (I would have been astounded if they had given the range of potentially confounding factors), the simulations presented the authors provide a very informative exploration of the sensitivity of this system to different climate forcing scenarios. The findings of this paper have implications for others aiming to simulate ice sheet evolution in topographically complex regions, both in terms of palaeo-simulations and contemporary scenarios/projecting future change, though the latter could come through more clearly especially in the conclusion.

On a personal note I thoroughly enjoyed reading this paper, and I have very few substantive comments to make on the science. Having done a lot of fieldwork in this region it's great to see this work done, as I've often wondered how you would even attempt to go about effectively modelling the entire fjord system over these timescales given challenges of resolution, computational cost, boundary conditions, and model physics. In this paper Cuzzone et al. deal with each of these issues in the most robust way that is currently feasible, and are able to provide valuable insights into the controls on Holocene deglaciation of this region.

My only substantive comment on the manuscript is primarily stylistic, in that parts of the results section occasionally stray into discussion (e.g. L322-8; L334-6 and others), while L396-405 does not really fit in this section. What is there is important, and should not be removed from the paper, so I would ask that the authors go through this section and pull out any interpretation of results and reallocate them to the discussion.

As alluded to previously, I think the authors currently undersell the relevance of this work for those working on the deglaciation of other topographically complex regions (in palaeo/contemporary/future contexts), and it would be nice to see this come through a bit more clearly in the introduction, relevant parts of the discussion and conclusion especially.

James Lea

**Minor points**

L26 – current best practice on place names is to provide Greenlandic name, followed by Danish colonial name in brackets e.g. Nuup Kangerlua (Godthåbsfjord)

L40 – if implementation and resolution of calving is important for robust past simulations, will it not also be important for contemporary simulations/future decadal to centennial projections too?

L59 – interglacial rather than interglaciation?

L225 – 8.3 mm/deg C/day?

L248-9 – why is 40 m/yr chosen?

Section 3.4 – a sentence on how the model domain in the ice sheet interior was defined would be useful. Obviously any upstream impacts on flux will be partially mitigated by the domain boundary condition, though given the model is forced in large part by T and precipitation there are potential downstream impacts on having the domain defined as is (given that the contemporary upstream catchment of KNS extends further to the south). To be clear, I do *not* think this undermines any of the results in the paper – defining palaeo-catchments for ice sheet outlets is tricky to impossible ab initio. However a sentence or two on why this is not a huge issue for the results would be useful for the reader.

Section 3.4 – is GIA accounted for? I know this region is pretty complicated in terms of it's GIA response, though a sentence on the expected range of bedrock change and how this may/may not impact results (particularly impacts on calving) would be worth flagging here.

L264 – missing bracket

L402-405 – this is perhaps a misunderstanding on my part, but I do not think that the assertion that other published simulations that show retreat inboard of the present day ice margin are likely too extreme (L402-405) is fully substantiated by results presented, *unless* the authors are referring to the scale of retreat relative to the present ice margin. There is evidence for terrestrial portions of the ice sheet being inland of the current ice margin along the SW coast during the Holocene thermal maximum (e.g. Larsen et al., 2015 [https://doi.org/10.1130/G36476.1]; and referred to by the authors L429-426), and while I fully acknowledge the differences between terrestrial and marine terminating margins, I still think the assertion made (as written) goes too far.

Figure 11 – can you change the colour map here to something other than red/green as it's a bit challenging for a colour blind person to see!

L633 – fjord rather than ford

L658-675 – The majority of this paragraph reads as a bit of a list currently, and think it would be good to expand a little (half a sentence or so) on each point on their wider implications (both for KNS region and generally across Greenland). This would bring the paper into line with sentences mentioned in the abstract (e.g. L37-40)

---

## Author Comment (AC1)

We would like to thank the reviewer's (James Lea and Anonymous Referee#2) for their constructive review and feedback on our paper. Below you will find our response (in black text) to the reviewer comments (in blue text) and the details regarding changes made to the paper following their recommendations.

Reviewer #1, James Lea

- In this paper Cuzzone et al. present a series of simulations of the Nuup Kangerlua (Godthåbsfjord) region, driven primarily by novel paleoclimate simulations of temperature and precipitation. Within these, they test high/low temperature/precipitation scenarios and scenarios for each where a levelset calving criterion is turned on/off. Their results show surface mass balance to exert a strong control on rapid retreat (consistent with geological evidence), while calving primarily exerts a strong control on ice dynamics (and therefore the evolution of total domain volume that is arguably more important). While their simulations are not able to recreate the contemporary ice margin (I would have been astounded if they had given the range of potentially confounding factors), the simulations presented the authors provide a very informative exploration of the sensitivity of this system to different climate forcing scenarios. The findings of this paper have implications for others aiming to simulate ice sheet evolution in topographically complex regions, both in terms of paleo-simulations and contemporary scenarios/projecting future change, though the latter could come through more clearly especially in the conclusion. On a personal note, I thoroughly enjoyed reading this paper, and I have very few substantive comments to make on the science. Having done a lot of fieldwork in this region it's great to see this work done, as I've often wondered how you would even attempt to go about effectively modelling the entire fjord system over these timescales given challenges of resolution, computational cost, boundary conditions, and model physics. In this paper Cuzzone et al. deal with each of these issues in the most robust way that is currently feasible, and are able to provide valuable insights into the controls on Holocene deglaciation of this region.

Thank you for the thorough review. It is encouraging to receive positive feedback on our experimental setup and execution, particularly given the reviewers expertise in this region of Greenland.

> My only substantive comment on the manuscript is primarily stylistic, in that parts of the results section occasionally stray into discussion (e.g. L322-8; L334-6 and others), while L396-405 does not really fit in this section. What is there is important, and should not be removed from the paper, so I would ask that the authors go through this section and pull out any interpretation of results and reallocate them to the discussion.

We worked to get rid of any discussion from the results section following the reviewer recommendation. For Lines 322-338, we realized that some of this text was actually discussed in the Discussion section 5.1. So, we removed most of the text that was previously occupying L322-328.

For Lines 334-336, we moved this sentence to the Discussion section 5.1 Line 582 where we discuss ice surface lowering simulated by the ice model.

- As alluded to previously, I think the authors currently undersell the relevance of this work for those working on the deglaciation of other topographically complex regions (in palaeo/contemporary/future contexts), and it would be nice to see this come through a bit more clearly in the introduction, relevant parts of the discussion and conclusion especially.

We tried to address these concerns throughout the text. In particular we added a statement about typical paleo ice flow model resolution in the introduction:

"However, as many paleo ice flow models employ model grids that are relatively coarse (10 km or greater), ice margin migration and ultimately ice discharge through fjord systems may be poorly simulated or not captured as many models cannot resolve the complex and narrow fjord geometries found across the GrIS (Cuzzone et al., 2019)."

And in the Discussion and Conclusion we have added text, which may also be relevant to our response to your last major comment (see below).

- L26 – _current best practice on place names is to provide Greenlandic name, followed by Danish colonial name in brackets e.g. Nuup Kangerlua (Godthåbsfjord)

Thank you for informing us on this. We have changed the text to reflect your recommendation.

- L40 – _if implementation and resolution of calving is important for robust past simulations, will it not also be important for contemporary simulations/future decadal to centennial projections too?

Yes, we do think that these processes are important for contemporary and future GrIS mass evolution. We have changed the text (Line 37) to reflect a more comprehensive view as recommended by the reviewer as follows: "While these results imply that the implementation and resolution of ice calving in paleo ice flow models is important towards making more robust estimations of past ice mass change, they also illustrate the importance these processes have on contemporary and future long term ice mass change across similar fjord-dominated regions of the GrIS. "

- L59 – _interglacial rather than interglaciation?

Changed text to "interglacial"

- L225 – _8.3 mm/deg C/day?

Yes, thanks for pointing that out. We have updated it to the correct units ($mm^{-1}ºC^{-1}day^{-1}$).

- L248-9 – _why is 40 m/yr chosen?

We chose this value as it is close to (or within a range of) melt rate values derived for contemporary floating ice shelves across the GrIS from Wilson et al., 2017. We have updated the text (Line 209) as follows, and added the citation listed below: "For these simulations, we apply a temporally constant melting rate under floating ice of 40 m/yr, which is consistent with contemporary melt rates derived near the grounding line of floating ice shelves across the GrIS (Wilson et al., 2017)."

Wilson, N., Straneo, F., and Heimbach, P.: Satellite-derived submarine melt rates and mass balance (2011–2015) for Greenland's largest remaining ice tongues, The Cryosphere, 11, 2773–2782, https://doi.org/10.5194/tc-11-2773-2017, 2017.

- Section 3.4 – _a sentence on how the model domain in the ice sheet interior was defined would be useful. Obviously any upstream impacts on flux will be partially mitigated by the domain boundary condition, though given the model is forced in large part by T and precipitation there are potential downstream impacts on having the domain defined as is (given that the contemporary upstream catchment of KNS extends further to the south). To be clear, I do *not* think this undermines any of the results in the paper – _defining palaeo-catchments for ice sheet outlets is tricky to impossible ab initio. However a sentence or two on why this is not a huge issue for the results would be useful for the reader.

This is a great point. We agree that paleo changes in the catchment for the KNS will impact the ice flux across our domain. As the reviewer pointed out, because these paleo changes in the catchment extent are not constrained, we have wiggle room in how we determine our model domain and its extent. In order to justify and give confidence to our model domain choice, we have added text as follows:

"The eastern boundary of our model domain extends outward to the present-day ice divide (Rignot and Mouginot, 2012), with the northern and southern boundary of our model domain extending to cover the KNS forefield. While the catchment for KNS may have changed during the Holocene and thus may have impacted ice flux into our domain, those changes are not constrained. Therefore, since we use consistent boundary conditions across our experiments, we consider that our results are primarily influenced by the surface climate and oceanic boundary conditions applied and not influenced by model domain extent."

- Section 3.4 – _is GIA accounted for? I know this region is pretty complicated in terms of it's GIA response, though a sentence on the expected range of bedrock change and how this may/may not impact results (particularly impacts on calving) would be worth flagging here.

GIA is not accounted for in these simulations. During the Holocene, crustal rebound could have been on order 100 or more meters in total (including restrained rebound; using output from Caron et al., 2018). This would likely have a minor influence for our model domain with respect to calving glaciers, but we don't think this particular influence would significantly impact rates of mass loss across the model domain. Elevation changes of the ice surface due to 10's of meter rise in the underlying bedrock topography as the ice mass responded to Holocene climate change would likely have a minimal impact on the surface mass balance. To address this limitation, we have added (Line 299):

"Discussed further in Section 5.3, we do not include glacial isostatic adjustment (GIA) in these simulations. Although GIA can influence the underlying bedrock topography and ultimately surface mass balance gradients and grounding line stability, changes during the Holocene across our domain are likely small (i.e. on the order of 100 meters; Caron et al., 2018), and therefore we expect this to have a minimal impact on our simulated ice histories."

- L264 – _missing bracket

Thanks, we have corrected this.

- L402-405 – _this is perhaps a misunderstanding on my part, but I do not think that the assertion that other published simulations that show retreat inboard of the present day ice margin are likely too extreme (L402-405) is fully substantiated by results presented, *unless* the authors are referring to

the scale of retreat relative to the present ice margin. There is evidence for terrestrial portions of the ice sheet being inland of the current ice margin along the SW coast during the Holocene thermal maximum (e.g. Larsen et al., 2015 [https://doi.org/10.1130/G36476.1]; and referred to by the authors L429-426), and while I fully acknowledge the differences between terrestrial and marine terminating margins, I still think the assertion made (as written) goes too far.

Apologies for the miscommunication in our text, but we are indeed referring to the 'scale' of inland ice margin retreat relative to present day ice margin.  To make this clearer we have adjusted the text as follows:

"Although no direct geological constraints on the minimum GrIS ice extent during the Holocene exist, available constraints suggest that the magnitude of large-scale ice margin retreat inboard of the present-day extent as simulated by some ice sheet models in this sector (20-40 km; Tarasov and Peltier, 2002; Lecavalier et al., 2014) is likely too extreme."

- Figure 11 – _can you change the colour map here to something other than red/green as it's a bit challenging for a colour blind person to see!

  Yes, we have adjusted the colors here to be more colorblind friendly.  We ended up keeping things consistent and use the same colormap as in Figs 5,6 and 9,10.

- L633 – _fjord rather than ford

Thanks.  The change has been made.

- L658-675 – _The majority of this paragraph reads as a bit of a list currently, and think it would be good to expand a little (half a sentence or so) on each point on their wider implications (both for KNS region and generally across Greenland). This would bring the paper into line with sentences mentioned in the abstract (e.g. L37-40).

  Thank you.  We approached these changes by trying to expand upon the role of capturing ice discharge in paleo ice flow models, with goals to improve estimate of past ice mass change.

  "Our modeling results shed light on the well constrained observations of Holocene ice retreat across the KNS forefield.  These simulations agree well with observations that ice retreat on terrestrial bedrock surfaces occurred rapidly between ~11.5 ka to 9.5 ka in response to early Holocene warming. The variations in the timing and magnitude of ice retreat on terrestrial bedrock surfaces across this region are found to be insensitive to calving within the fjords that intersect this landscape.  Instead, the terrestrial ice retreat is more sensitive to SMB, with warmer climate reconstructions providing the best fit between the modeled and observed ice retreat.  Calving, however, does play a significant role in the simulated Holocene ice volume change across this domain.  Acting as conduits for mass transport and ice flux, ice velocity within the fjords in the KNS forefield increases when the ice front is allowed to calve.  Calving promotes further ice mass transport from the interior of the domain to the ice front which helps to thicken ice within the fjords, allowing the ice front to persist longer than adjacent terrestrial margins similar to the ice response simulated for the Holocene retreat of Jakobshavn Isbræ (Kajanto et al., 2020).  These results suggest that paleo ice-flow models that do not sufficiently resolve fjord geometry may not capture dynamic processes that are critical towards understanding long term ice mass change across the GrIS.  Recent ice flow modelling has suggested

that despite increased ice mass loss due to a more negative SMB, ice discharge from GrIS marine terminating glaciers will play a significant role in overall GrIS mass change well into the future (Choi et al., 2021).  These results confirm that over paleoclimate timescales, while SMB may dictate large scale ice margin migration as captured in geologic observations, ice discharge can greatly influence the rate and magnitude of ice mass change.  However, as all simulations depict contemporary ice extent that is too extensive, uncertainties in the reconstruction of past climate and model parametric uncertainties ultimately contribute to misfits that are difficult to quantify given our computationally expensive model setup.  Future paleoclimate ice flow modelling with ISSM will aim to take advantage of recent advances in statistical emulation (e.g., Edwards et al., 2021) to better quantify the influence of model parametric uncertainty on simulated Holocene ice retreat. "

---

## Author Comment (AC2)

We would like to thank the reviewer's (James Lea and Anonymous Referee#2) for their constructive review and feedback on our paper. Below you will find our response (in black text) to the reviewer comments (in blue text) and the details regarding changes made to the paper following their recommendations.

Reviewer #2, Anonymous Referee

Cuzzone et al. present have simulated the deglaciation of the GodthaÌ bsfjord region using different climate forcings (temp, precip.) including a calving laws. Their results show that the Holocene deglaciation was primarily dominated by changes in surface mass balance, whereas calving is less important. The simulations are compared to geological reconstructions of the deglaciation. Overall, the paper reads well and presents some new and interesting aspects on how to simulate ice sheet evolution (as far as I can tell as a none-expert in ice sheet modelling). I only have a few major comments and some additional minor comments to the manuscript.

Thank you for your review of our work. Below we address your concerns and identify in text where changes have been made.

Major comments:

I am surprised that it is not quantified how much of the deglaciation is forced by SMB and calving. Most places it is vague formulated like "significant influence", "strongly impacted by" or "less important contribution from". Is it possible to be more specific i.e. say xx% from SMB and yy% from calving?

We understand the concern and interest in trying to partition the relative contribution to the deglaciation from SMB and iceberg calving. Unfortunately, this is very difficult to do as the SMB and calving are interconnected. For example, as the ice surface elevation changes in response to SMB or calving, the ice dynamics will change, impacted calving and SMB alike.

Our experimental setup and implementation instead allowed us to capture the influence of iceberg calving on the deglaciation across this domain in a relatively simple manner. By running the transient simulations with and without the iceberg calving turned on, we can directly compare the simulated deglaciation ages. These experiments conclude that iceberg calving played a negligible role on the deglaciation across the KNS forefield as captured by the geologic constraints.

It is unclear to me if they account for changes in sea surface temp in the fjords? Changing SST should influence the calving rate, but it is unclear to me if this is accounted for in the model. It is known from paleoclimate data that the SST changes significantly during the Holocene and this could have played in significant role on the ice retreat in particularly in fjord settings like the GodthaÌ bsfjord region (see review by Axford et al2021).

While SST records exist and indicate variations across the Holocene, the ice front would be more susceptible to changes in sub-surface ocean temperatures, of which data is limited and lacking. Likewise, estimating basal melt rates is difficult although approaches are being implemented in ice flow modelling currently. Because of these limitation in data, we did not include variation in ocean temperature, and discuss this in Section 5.3 (Model Limitations). We added an additional sentence (Line 647: "While proxy

records indicate changing sea surface temperatures during the Holocene proximal to our model domain Axford et al. (2021), due to a lack of constraints on the long-term subsurface ocean thermal forcing needed to implement undercutting in our simulations, we opted to disregard this."). We also discuss here that although we do not include the influence of melt rates on the vertical calving face of the ice front, we adjust the von misses stress threshold to a lower value (600 kPa) which would likely account for the possible role of oceanic melt and undercutting. While lowering this threshold showed no appreciable influence on the deglaciation or mass loss, it is not absolutely conclusive that the ocean and variations in sub-surface temperature played a negligible role. We are currently addressing this as part of a much larger project, where we use a model PICOP coupled to ISSM (Pelle et al., 2019) to estimate melt rates from far field temperature and salinity changes.

To this end we included this statement in the limitations (Line 656: "Future work will use a basal melt-rate parametrization (PICOP; Pelle et al. 2019), employed in ISSM currently, to estimate oceanic melt rates from far field variations in Holocene subsurface temperature and salinity in order to more robustly estimate the impact of oceanic warming Holocence deglaciation across the GrIS.")

Pelle, T., Morlighem, M., and Bondzio, J. H.: Brief communication: PICOP, a new ocean melt parameterization under ice shelves combining PICO and a plume model, The Cryosphere, 13, 1043–1049, https://doi.org/10.5194/tc-13-1043-2019, 2019.

The 9.3 ka and 8.2 ka re-advance events have been recorded north of the study areas around Jakobshavn Isbræ (e.g. Young et al2011). In the GodthaÌ bsfjord region there is no evidence of a readvance during these two cold events – neither in the geological data or in the simulations (as far as I can tell from the figures). It would be interesting if it could be discussed why the ice sheet in this GodthaÌ bsfjord region did not react to these events. One possibility is that the ice sheet retreated far inland during the Early Holocene and that the 9.3 and 8.2 ka re-advances was minor and did not pass the LIA extent. However, according to the simulations the ice margin did not retreat far inland of the present extent. How could this be explained?

Figure 7 (Ice volume) shows a pause or slight increase in ice volume during the 9.3 ka and 8.2 ka event. We just want to reiterate that these are simulated ice margins from the model experiments, not necessarily what occurred in reality. Thus, slight pauses or increases in ice volume at 9.3 and 8.2 ka indicate that our model is capturing/responding to the 9.3 and 8.2 ka events, but the ice-margin response is rather limited. Thus, the simulations do show a response to these events, and the ice margin (not shown in the paper plots) does stabilize and in some location's advances (although very minor).

The reviewer is correct in that in the immediate KNS region, geologic constraints reveal that the ice margin retreated behind the eventual historical maximum position between ~10.3 and 9.5 ka (Fig. 2) so therefore any expression of the 9.3 and 8.2 ka events would not be preserved on that landscape. Rather, the LIA/historical advance overran any potential 9.3 or 8.2 ka moraines

Line 26 delete "novel"

done

Line 29 change bedrock with fjord

We decided to change it to 'terrestrial' to indicate land above sea level.

Line 29 what do you mean with "above sea level"?

See above

Line 34 and throughout the manuscript: capitalize "early", "middle" and "late" Holocene.

done

Line 47: capitalize smb and use consistently in MS

done

Line 63, 86: (s)outhwestern

done

Line 264: ) is missing.

done

Line 287: . is missing after et al

done

Table 1 is not really needed.

We prefer to leave this in as a reminder for the set of experiments needed, but are willing to remove if it is absolutely needed.

Line 481: Format reference.

done

Line 630: Both "Isbræ" and "glacier" have been used.

done

Line 633: change "ford" to fjord.

done

Line 642-656: Mostly not conclusions and could be omitted.

Perhaps stylistic, but this paragraph was intended to set a tone for the conclusions that follow.  Again, we would prefer to keep this, but are fine omitting if the reviewer/editor find it absolutely necessary.

Line 741-744: check format

done

Line 783-786: check format

done

Line 793-802: check format
done